# Low iron-induced small RNA BrrF regulates central metabolism and oxidative stress responses in *Burkholderia cenocepacia*

**Andrea M. Sass**, **Tom Coenye** *

Laboratory of Pharmaceutical Microbiology, Ghent University, Ghent, Belgium

* tom.coenye@ugent.be

## Abstract

Regulatory small RNAs play an essential role in maintaining cell homeostasis in bacteria in response to environmental stresses such as iron starvation. Prokaryotes generally encode a large number of RNA regulators, yet their identification and characterisation is still in its infancy for most bacterial species. *Burkholderia cenocepacia* is an opportunistic pathogen with high innate antimicrobial resistance, which can cause the often fatal cepacia syndrome in individuals with cystic fibrosis. In this study we characterise a small RNA which is involved in the response to iron starvation, a condition that pathogenic bacteria are likely to encounter in the host. BrrF is a small RNA highly upregulated in *Burkholderia cenocepacia* under conditions of iron depletion and with a genome context consistent with Fur regulation. Its computationally predicted targets include iron-containing enzymes of the tricarboxylic acid (TCA) cycle such as aconitase and succinate dehydrogenase, as well as iron-containing enzymes responsible for the oxidative stress response, such as superoxide dismutase and catalase. Phenotypic and gene expression analysis of BrrF deletion and overexpression mutants show that the regulation of these genes is BrrF-dependent. Expression of *acnA*, *fumA*, *sdhA* and *sdhC* was downregulated during iron depletion in the wild type strain, but not in a BrrF deletion mutant. TCA cycle genes not predicted as target for BrrF were not affected in the same manner by iron depletion. Likewise, expression of *sodB* and *katB* was dowregulated during iron depletion in the wild type strain, but not in a BrrF deletion mutant. BrrF overexpression reduced aconitase and superoxide dismutase activities and increased sensitivity to hydrogen peroxide. All phenotypes and gene expression changes of the BrrF deletion mutant could be complemented by overexpressing BrrF *in trans*. Overall, BrrF acts as a regulator of central metabolism and oxidative stress response, possibly as an iron-sparing measure to maintain iron homeostasis under conditions of iron starvation.

## Introduction

*Burkholderia cenocepacia* J2315 is a member of the *Burkholderia cepacia* complex (BCC), a group of aerobic Gram-negative beta-proteobacteria which mainly live in the rhizosphere, but can also act as opportunistic pathogens, particularly in individuals with cystic fibrosis [1].

**Data Availability Statement:** All relevant data are within the manuscript and its Supporting Information files.

**Funding:** TC received funding from the Belgian Science Policy Office (Belspo) through the Interuniversity Attraction Pole Program (Phase VII/2012–2017, Project P7/28). http://www.belspo.be/belspo/iap/index_en.stm The funders had no role in study design, data collection and analysis, decision to publish, or preparation of the manuscript.

**Competing interests:** The authors have declared that no competing interests exist.

Iron is essential for living organisms, as part of the catalytic/active site of many enzymes, yet it is mostly inaccessible for bacteria due to the low solubility under oxic conditions at neutral pH [2, 3]. Iron acquisition mechanisms are therefore important for survival of bacteria [4], as well as mechanisms to conserve iron under conditions of iron depletion. On the other hand, higher intracellular concentrations of iron are toxic to cells [3]. Free iron can react with the superoxide anion and with $H_2O_2$, natural by-products of respiration and oxidase reactions, to form the very reactive hydroxyl radical via the Fenton reaction. All reactive oxygen species (ROS) can damage DNA, proteins and lipids, and an excess of ROS leads to cell death. For protection against ROS, cells produce detoxifying enzymes such as catalases and superoxide dismutases (SOD) [3].

The ferric uptake regulator (Fur) is important for iron homeostasis in many bacteria. Its best known mechanism of action is the repression of genes involved in iron uptake under iron replete conditions by iron-dependent binding to a specific sequence motif, the Fur box [2]. In *Escherichia coli* and *Pseudomonas aeruginosa* Fur also positively regulates SOD, the iron scavenger protein bacterioferritin and several enzymes of the tricarboxylic acid cycle (TCA) [5, 6]. This positive regulation was attributed to indirect effects mediated by Fur-regulated small RNAs (sRNAs), RyhB in *E. coli* [6] and PrrF in *P. aeruginosa* [5]. Under iron-depletion the Fur repression of these sRNAs is lifted, they bind to the mRNA of their targets and the sRNA-mRNA hybrid is then rapidly degraded, or the translation of targets is inhibited [7]. This reduces the demand for iron in the cell, since many targets have iron or iron-sulfur clusters as cofactor.

A small RNA highly upregulated under iron depletion was identified in *B. cenocepacia* J2315 by screening dRNA-Seq data for short transcripts [8, 9] and designated ncS63. Its computationally predicted targets included confirmed targets of *E. coli* RyhB and *P. aeruginosa* PrrF, such as *sdhC* and *sodB* [5, 6, 10, 11], yet ncS63 has no sequence similarity to any known Fur-regulated sRNAs. Here we report on the full characterisation of this low iron-induced sRNA. As our results suggest that *B. cenocepacia* ncS63 shows analogy to RyhB and PrrF, we suggest the name BrrF (*Burkholderia regulatory RNA involving iron, Fe*), analogous to *P. aeruginosa* PrrF [5] and *Neisseria meningitidis* NrrF [12].

## Materials and methods

### Media

Strains were routinely cultured in LB broth or agar (low-salt Lennox formulation: 10 g/L tryptone, 5 g/L yeast extract, 5 g/L NaCl, 1.5% agar), supplemented with 600 μg/ml trimethoprim (TP) as selective antibiotic where appropriate. Gene expression from the plasmid was induced by adding rhamnose (Sigma) to a final concentration of 0.2% (w/v). Iron depletion was induced by adding the iron chelator 2,2'-dipyridyl (Sigma) at 200 μM final concentration to LB broth or agar.

For growth curves on single carbon compounds, a phosphate buffered mineral medium (basal salt medium, BSM) was used; which contained 18.6 mM $K_2HPO_4$, 7.2 mM $NaH_2PO_4$, 37.4 mM $NH_4Cl$, 0.1 g/L nitrilotriacetic acid, 0.2 g/L $MgSO_4$ x 7 $H_2O$, 0.012 g/L $FeSO_4$ x 7 $H_2O$, 0.003 g/L $MnSO_4$ x $H_2O$, 0.003 g/L $ZnSO_4$ x 7 $H_2O$, 0.001 g/L $CoSO_4$ x 7 $H_2O$, [13]. As *B. cenocepacia* J2315 does not grow in BSM when iron is depleted by addition of dipyridyl or by treating the medium with Chelex, low iron conditions were obtained by omitting iron sulfate.

All incubations were performed at 37˚C.

### Strains and plasmids

*B. cenocepacia* strain J2315 (S1 Table) was used as background for all experiments. A *brrF* deletion mutant was constructed by allelic recombination as described by Aubert et al. [14].

Deletion of *brrF* was confirmed by Sanger sequencing of a 449 nt long region spanning *brrF*. Amplification of all inserts was performed with the HotStar HiFidelity Polymerase Kit (Qiagen).

For overexpressing BrrF, a region ranging from 80 nt upstream of the putative processing site to 49 nt downstream of the computationally predicted terminator sequence was amplified and cloned into an overexpression vector with a rhamnose-inducible promoter [15], yielding pBrrF-d1. The overexpression vector had previously been modified to remove the ribosome binding site and start codons [16].

The transcript expressed from pBrrF-d1 needs to be processed/cleaved to produce the native BrrF. Most computationally predicted interactions with targets were located at the very 5'end of the cleaved transcript, making it the most interesting region to introduce point mutations for complementation experiments. Introducing mutations so close to the cleavage site could potentially interfere with the processing of the overexpression transcript. For this reason, a second overexpression vector was constructed by inverse PCR with pBrrF-d1 as template, removing the sequences upstream of the putative processing site, and yielding pBrrF-d2. In the same manner, a vector carrying a *brrF* derivative with point mutations at position 8 and 9 was constructed, also by inverse PCR with pBrrF-d1 as template, yielding pBrrF-d3.

Overexpression plasmid derivatives were created by inverse PCR with the LongRange PCR kit (Qiagen). PCR products were digested with DpnI (Promega) to remove template plasmid, blunted with T4 DNA polymerase (Promega) to remove A-overhangs and gel-cleaned. Amplified plasmid was then phosphorylated using T4 polynucleotide kinase (Promega) and self-ligated (T4 DNA ligase, Promega). All plasmid inserts were verified by Sanger sequencing. Plasmids were transformed into *B. cenocepacia* J2315 by triparental mating and expression of BrrF from the plasmid in the presence of rhamnose was confirmed by qPCR.

All primer sequences are listed in S2 Table.

## Growth curves

Growth on single carbon sources was measured in round-bottom 96-well microtiter plates (SPL Life Sciences) in a temperature-controlled microplate reader (Envision, Perkin Elmer), with intermittent shaking. Cells were inoculated at $10^6$ CFU/ml and optical density (O.D.) was measured at 30 min intervals over >60 hours. The mineral medium was supplemented with 0.05% yeast extract and 0.05% casamino acids, without which strain J2315 would grow extremely slowly on only one carbon source. Controls with only the organic supplements were run on every plate. Compounds were purchased from Sigma-Aldrich, Fluka, Janssen, and Acros Organics. Every compound was tested in at least two independent biological replicates.

Growth in LB broth was investigated using glass flasks in a shaking incubator. Growth was monitored via backscattered light every 2 min using a Cell Growth Quantifier (CGQ, Aquila Biolabs).

Growth rates were determined using the equation $\mu * h^{-1} = \ln(x_{t2}/x_{t1})/(t_2-t_1)$, where x denotes optical density or backscatter arbitrary units, and $t_1$ and $t_2$ refer to a point at the beginning or end of the analysed time interval, respectively [17]. The obtained values for μ of the biological replicates were then analysed either by One-way ANOVA or a two-tailed Student's t-test using SPSS (v. 25).

## Sensitivity to $H_2O_2$

Cell suspensions with an O.D. (595 nm) of 0.1 (approx. $10^8$ colony forming units (CFU) per ml) were spread on low-nutrient LB agar plates (1 g/L tryptone, 0.5 g/L yeast extract, 5 g/L NaCl, 1.5% agar) with a sterile cotton swab. Inoculated plates were pre-incubated for 1 hour to

allow gene expression to start. Then, 10 μl of a 1% $H_2O_2$ (Sigma) solution were applied to 6 mm blank susceptibility testing discs (Oxoid) which were placed into the middle of the agar plate. Tests were performed in six replicates, on three different days, and growth inhibition zones were measured after 24 h incubation.

## Superoxide dismutase activity measurements

SOD activity was determined with a SOD assay kit (Sigma 19160-1KT-F), which uses a water soluble tetrazolium salt that reacts with superoxide produced by a xanthine oxidase. SOD inhibits this reaction by removing the superoxide. Cells were grown in 25 ml LB broth in flasks on a shaker as described above, without addition of inducers, to an O.D. of 0.5 ($5 \times 10^8$ CFU/ml). Then the inducer was added and incubation continued for another hour (pulse expression). Cultures were then cooled in ice water and centrifuged at 4°C. The cells were washed twice in 25 ml 50 mM ice-cold sodium phosphate buffer, pH 7, re-suspended in 1 ml of the phosphate buffer supplemented with 1 mg/ml dithiothreitol, and centrifuged again. The supernatant was removed and the cell pellet stored at -80°C for a maximum of 24 hours. Cells were lysed in 1 ml ice cold phosphate buffer by bead beating with zirconia-silica beads (0.1 mm diameter, BioSpec). The lysate was centrifuged at max speed for 5 min at 4°C, the supernatant transferred into a fresh tube and kept on ice. The SOD assay was performed immediately according to the manufacturer's instructions. The protein concentration in the extract was determined with a Bradford assay (Sigma) and used for normalisation of the SOD activity.

## Aconitase activity measurements

Aconitase activity was determined by measuring the production of *cis*-aconitate. To this end isocitrate was added to a crude cell extract in the presence of high concentrations of citrate and the increase of absorbance at 240 nm was measured [18–20].

Cells were grown in flasks as mentioned for SOD activity measurements. After harvest (as above), cell were washed twice in a TRIS/citrate buffer (20 mM citric acid, 20 g/L TRIS, pH 8), resuspended in 1 ml ice cold TRIS/citrate buffer and immediately lysed by bead beating. The lysate was centrifuged and the supernatant kept on ice. For aconitase activation, freshly prepared cysteine and ammonium iron (II) sulfate solutions were subsequently added to a final concentration of 5 and 0.5 mM, respectively. Supplemented lysates were then incubated on ice for one hour before measurements.

DL-isocitrate ($Na_3$-salt, Sigma) was added to the extract to a final concentration of 40 mM, at room temperature in a Biodrop μLite spectrophotometer, with a microvolume sample port of 0.5 mm path length. The absorbance at 240 nm was measured in intervals of 5 seconds for a total of 5 min and corrected for absorbance of the same sample without isocitrate addition. The protein concentration in the extract was determined with a Bradford assay, and a standard curve was generated with *cis*-aconitate (Sigma) added to heat-inactivated lysates (supplemented lysates heated to 50°C for 15 min and centrifuged) containing 40 mM isocitrate.

## qPCR

Cells were grown in 25 ml LB broth in flasks on a shaker as described above. Rhamnose and/or dipyridyl were added 30 min before flasks were cooled in ice water and 4 ml culture harvested by centrifugation in microcentrifuge tubes at 4°C and maximum speed. This "pulse-expression" method was chosen to minimise secondary effects of iron depletion or BrrF-overexpression on target gene expression. Pellets were stored at -80°C for a maximum of one week. RNA was extracted with the RiboPure bacteria kit (ThermoFisher) with the following changes to the manufacturer's instructions: the crude extract was mixed with 1.25 x volumes ethanol before

column cleaning to better retain sRNAs, and DNase I treatment time was increased to 60 min. RNA was transcribed to cDNA with the High Capacity cDNA RT kit (Applied Biosystems). qPCR was performed as described before [9], using the GoTaq qPCR Master Mix (Promega). Each reaction was run with two technical replicates, each condition with 3 biological replicates and a no-RT control, and no-template controls were performed for each gene. Cq values were normalised to control gene *rpoD* (BCAM0918). One-Way ANOVA with a Tukey Post-hoc test using SPSS (v. 25) was performed to determine statistical significance. All primer sequences are listed in S2 Table.

## Computational methods

To identify loci with Fur boxes in the *B. cenocepacia* J2315 genome, and to predict the Fur box consensus sequence, a two-step procedure was employed. Fur box-containing upstream sequences of *B. multivorans* [21] were analysed using the MEME tool [22] and the resulting motif was submitted to FIMO [22] to identify Fur box-containing upstream sequences of genes induced under iron-depleted conditions according to a microarray reference dataset [23]. Of these, sequences comprising the entire 5'UTR and 150 nt upstream of the transcription start site (TSS) were extracted and returned to MEME for motif prediction. The resulting motif was then re-submitted to FIMO to screen for Fur boxes across the entire *B. cenocepacia* J2315 genome.

*brrF* homologs were screened for using BLASTn [24], with search parameters adjusted to short sequences with low similarity: word size 7, match/mismatch scores -1/1, gap costs 0/2 and expect threshold 0.001.

Global alignments of *brrF* homologs were generated with LocARNA [25], using default parameters (see S1 Fig for strains and sequences used). Secondary structures of BrrF and selected mRNAs were predicted with mfold [26] and visualised using StructureEditor [27].

BrrF target prediction was performed with CopraRNA [11], using default parameters and the *Burkholderia* strains from S1 Fig as input strains. This algorithm takes accessibility of inter-action sites and conservation of putative targets into account. Sequences adjacent to start codon of annotated genes, from 200 nt upstream to 100 nt downstream of the first nucleotide, were considered for target prediction; most confirmed sRNA-mRNA interactions are located in those regions. Predicted interactions located upstream of a known TSS [8] were removed from the results list. Functional enrichment analysis of predicted targets was performed with the DAVID tool [28], which is incorporated in CopraRNA.

## Results and discussion

### Sequence, conservation, genome context and regulation of BrrF

BrrF is located within the 3' end of BCAL2297, a small protein designated HemP for its involvement in heme iron uptake in *Burkholderia multivorans* (Fig 1A, [29]). BrrF appeared to be cleaved from the *hemP* mRNA, as indicated by the strong depletion of the BrrF transcript by treatment with a 5'-monophosphate-dependent exonuclease (TEX, [8]). Its size was determined as 126 nt by dRNA-Seq (position 2548559 to 2548684 on replicon 1), confirmed by Northern blotting [9]. BrrF does not extend into the computationally predicted rho-independent terminator (position 2548709 to 2548732) downstream of *hemP*, instead it terminates a at succession of U-residues without a preceding stem loop. The same genome context was identified for the BrrF homolog BTH_s39 in *Burkholderia thailandensis*, which was identified using tiling microarrays [30]. BTH_s39 is also located downstream of *hemP* and does not extend into its computationally predicted downstream rho-independent terminator. The size of BTH_s39 was determined by rapid amplification of cDNA ends and confirmed by Northern blotting, and is, with 130 nt, a close match to BrrF.

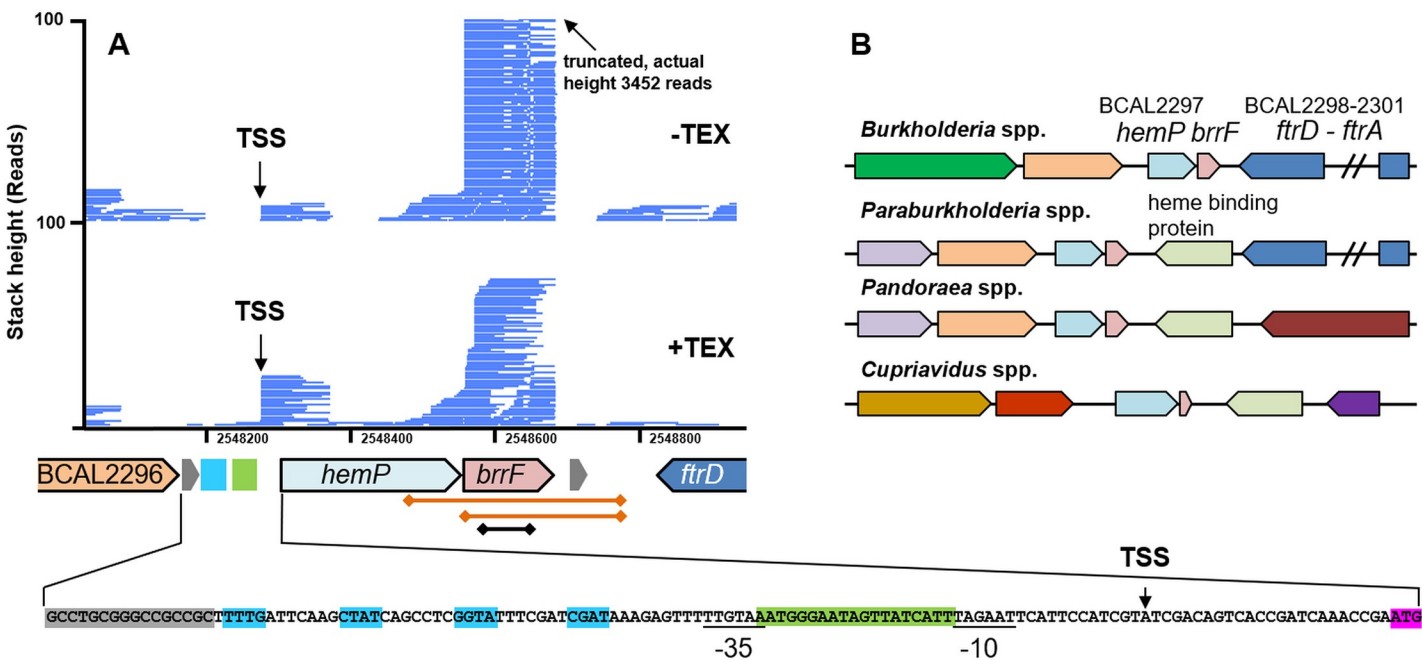

**Fig 1. Genome context and expression of *brrF*. A)** Coverage from differential RNA-Sequencing of a *B. cenocepacia* J2315 biofilm [8]. RNA samples were divided and one sub-sample treated with a 5'-monophosphate-dependent exonuclease (TEX), which selectively degrades processed transcripts. Reads starting at position 2548277 are enriched in the TEX-treated subsample, representing a transcription start site (TSS). Reads starting at position 2548559 are depleted by TEX-treatment, suggesting cleaving of the mRNA at this site. The *hemP* TSS (black arrow) is preceded by a 19 nt long Fur binding site (green box) and putative OxyR binding sites (blue boxes). Underlined: -10 and -35 box. Grey boxes: predicted rho-independent terminators. Pink box: *hemP* start codon. The diamond-ended lines below *brrF* depict the sizes of plasmid inserts for complementation (orange) and of the qPCR amplicon (black). **B)** Conserved genome context of *brrF*. On the upstream side, *brrF* is always flanked by *hemP* (light blue), downstream either by the *ftr* cluster for iron transport (dark blue) or a heme binding protein (light green). Identical colour denotes homologous proteins.

Although the 5'end of BrrF (GUAUU) bears similarities with typical RNase E consensus recognition sites in *E. coli* (RN↓WUU [31]), it is not cut at the central AUU as is typical for RNase E in that bacterium, but upstream of the first guanosine residue (CGA↓GUAUU, Fig 2A). The RNase recognition sequences in *B. cenocepacia* have not been systematically investigated, but the putative cutting sites of two other processed *B. cenocepacia* sRNAs (ncS35 CGA↓UUC [16], ncS27 CGA↓AUG [32]) are somewhat similar to the putative BrrF cutting site. This indicates that RNase recognition sites in *B. cenocepacia* differ from those in *E. coli*, and that one possible consensus recognition site might be CGA↓NUN.

*brrF* is conserved over the full length in the genera *Burkholderia* (100% coverage, 90–100% identity), *Paraburkholderia* (100% coverage, 81–89% identity) and in *Pandoraea* (100% coverage, 70–74% identity, S1 Fig). In particular the 5' end (nucleotides 4–45) shows a high degree of conservation. In the genus *Cupriavidus*, only nucleotides 4 to 76 are present (57% coverage, 75–80% identity). No similar sequences were found outside the order *Burkholderiales*. The genome context is also conserved, *brrF* is always located downstream of a *hemP* homolog. Located downstream of *brrF* is either the ferrous iron uptake operon FtrABCD [33] or a heme-binding protein (Fig 1B). The computationally predicted secondary structure of BrrF shows extensive internal base pairing, with two conserved hairpins at the 5' end (Fig 2A).

BrrF was previously found to be more than 50-fold up-regulated under iron-depleted conditions, whereas other tested growth conditions such as biofilm growth, oxidative stress, stationary phase and starvation did not induce BrrF expression [9]. *hemP* was upregulated by iron depletion to approximately the same fold change as *brrF*. HemP in *B. multivorans* is Fur regulated [21, 29]; therefore the Fur box consensus sequence for *B. cenocepacia* was

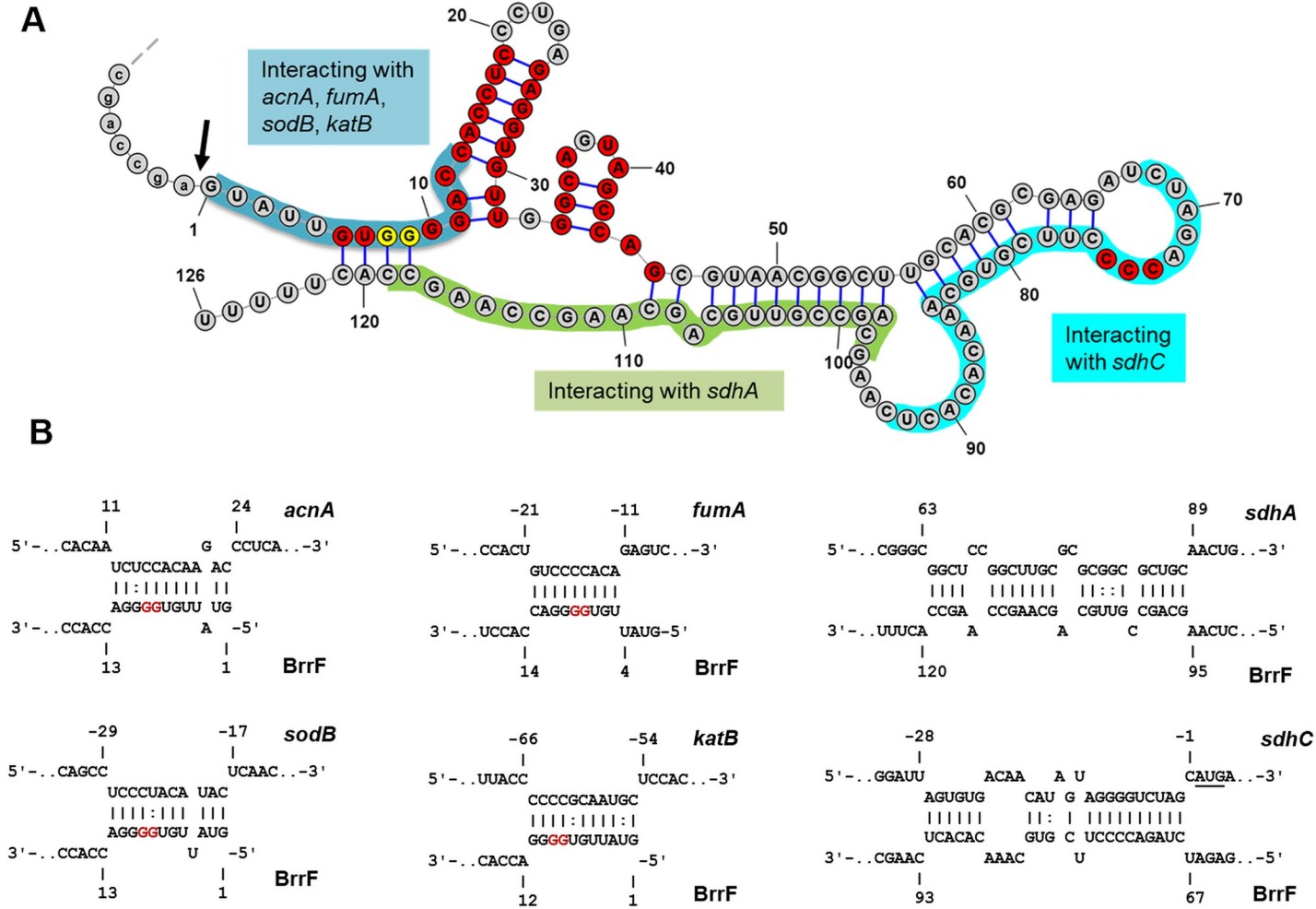

**Fig 2. Interactions of BrrF with mRNA targets. A)** Computationally predicted secondary structure of BrrF. Red bases are conserved across >95% of screened strains of genera *Burkholderia*, *Paraburkholderia*, *Pandoraea* and *Cupriavidus*. Yellow bases indicate the successive guanosine residues that were replaced by cytosine for overexpression experiments. Bases involved in predicted interactions with selected mRNA targets are shaded in colour. The black arrow indicates the putative cutting site. **B)** Computationally predicted interactions with mRNAs of *acnA*, *fumA*, *sdhC*, *katB*, *sodB* and *sdhA*. The numbers indicate nucleotide positions relative to the start of the coding sequence for the respective mRNA (upper sequence 5' to 3' from left to right) or the nucleotide position in BrrF (lower sequence, 5' to 3' from right to left). Nucleotide positions altered in overexpression experiments are in red.

determined (S2 Fig) and the whole genome screened for motif occurrences. Fur boxes were identified directly upstream of the TSS for *hemP* (Fig 1A), and upstream of known iron transport-related genes such as *orbS* and *fecI*. The sequence upstream of the *hemP* gene start in *B. cenocepacia* is identical to that upstream of *hemP* in *B. multivorans* for the first 77 nt, which includes the Fur box. This *B. multivorans* sequence was bound by *B. multivorans* Fur protein in an electrophoresis mobility shift assay [21], the respective sequence in *B. cenocepacia* is therefore likely also bound by Fur.

*brrF* itself did not have a Fur box directly upstream of its 5' end. Expression of BrrF is therefore probably under control of the Fur regulator via co-expression with *hemP*. In contrast to that, the Fur-regulated sRNAs RyhB in *E. coli* and PrrFs in *P. aeruginosa* are directly preceded by a Fur binding site.

*hemP* is possibly also under the regulation of OxyR. A putative OxyR box was found upstream of *hemP* (Fig 1A), and *hemP* was upregulated 10-fold in $H_2O_2$-treated biofilms of *B.*

*cenocepacia* J2315 [34] while in exponentially growing planktonic cells exposed to $H_2O_2$, neither *hemP* nor *brrF* change expression [9, 23]. However, in these studies biofilms were treated with 3% $H_2O_2$ for 30 min, whereas planktonic cultures were treated with 0.05 or 0.15% $H_2O_2$ for a shorter time period, which might account for the observed differences in *hemP* regulation under oxidative stress.

## Computationally predicted targets of BrrF

Target prediction resulted in 206 genes with sequence complementarity to BrrF (p-value $\leq$ 0.01, S3 Table). The categories with the largest number of genes with known function were amino acid transport and metabolism (12.6%) and energy production and conversion (10.0%). Most predicted interactions involved the 5' end of BrrF (S3 Table, S3 Fig), notable exceptions were targets *sdhC* and *sdhA* (Fig 2B). The predicted interaction regions in putative target mRNAs are located across the entire input region from 200 nt upstream to 100 nt downstream of the gene start (S3 Fig). Only a small proportion (17%) of the predicted mRNA interactions include the first 10 nt upstream of the gene start and the Shine-Dalgarno sequence, whereas most interactions were predicted for further upstream in the 5'UTR (38%) or within the coding sequence (45%, S3 Table). Secondary structure analysis of putative target mRNAs before and after virtual cleavage at the predicted interaction site did not indicate an occlusion of Shine-Dalgarno sequences by mRNA intramolecular structures. This indicates that binding of BrrF does not alter the rate of translation initiation, but that BrrF mainly acts by interfering with mRNA target degradation rate.

Functional enrichment analysis of predicted targets pointed to a relative over-representation of genes involved in aerobic respiration, iron- or heme-binding, and in the TCA cycle [9]. Furthermore, several genes involved in iron-sulfur cluster formation and in ROS detoxification, such as those encoding catalases and SOD, were predicted as targets (Table 1). These genes or functional categories are also targeted by RyhB and/or PrrF (see Table 1 for references).

The target prediction results and the similarity of BrrF to other Fur-regulated sRNAs suggests that BrrF is a regulator of the TCA cycle and of the oxidative stress response under conditions of iron depletion, most likely by down-regulating the putative target genes. Down-regulating genes encoding SdhC, SodB and other iron-containing proteins saves iron and facilitates iron homeostasis. Iron limitation and a decreased activity of TCA cycle enzymes leads to a decrease in ROS production, reducing the need for ROS detoxification. BrrF-dependent regulation could therefore also save energy and contribute to overall homeostasis in the bacterial cell.

A link between iron availability and expression of ROS detoxifying genes has previously been demonstrated in *B. multivorans*. A *B. multivorans* Fur mutant showed increased sensitivity to oxidative stress, reduced *sodB* expression and reduced SOD and catalase activities [21]; and also growth attenuation on many carbon compounds. These phenotypes could be complemented by deleting *hemP* together with the BrrF sequence from the *B. multivorans* genome [29].

## BrrF affects growth in *B. cenocepacia* J2315

The deletion mutant Δ*brrF* had no growth defect compared to wild type (WT) in iron-replete media, neither in a rich medium nor during growth on single carbon sources in a mineral medium (Fig 3A and S4 Fig). Δ*brrF* was also not impaired in heme utilisation, indicating that expression of HemP is not affected by *brrF* deletion (S4 Fig). Under iron limitation, Δ*brrF* grew marginally faster than WT. This is in contrast to observations with a PrrF double deletion

**Table 1. Regulation of computationally predicted BrrF target genes in response to iron depletion.**

| Predicted target | Gene | Annotation | Log2 fold change WT low Fe* | Log2 fold change ΔbrrF low Fe* | Ref.** |
|---|---|---|---|---|---|
| | | TCA and methylcitrate cycle | | | |
| BCAM0967 | sdhC | Heme-binding succinate dehydrogenase cytochrome b556 subunit | -1.28 | n.s. | [5, 10] |
| BCAM0969 | sdhA | Succinate dehydrogenase flavoprotein subunit | -2.25 | n.s. | [5, 10] |
| BCAM0961 | acnA | [Fe-S]-dependent aconitate hydratase A | -4.22 | n.s. | [10, 35] |
| BCAL2287 | fumA | [Fe-S]-dependent class I fumarate hydratase | -5.38 | 1.10 | [10, 35] |
| BCAL2908 | fumC | Class II fumarate hydratase, iron-free | 1.76 | 0.42 | [36] |
| BCAM2701 | acnM | [Fe-S]-dependent 2-methylisocitrate dehydratase | -2.00 | n.s. | |
| | | Defense against ROS | | | |
| BCAL2757 | sodB | $Fe^{2+}$-containing SOD | -2.33 | 0.75 | [5, 10] |
| BCAL3299 | katB | Bifunctional heme-containing catalase/peroxidase, homologous to *katG* in *E. coli* | -1.72 | 0.98 | [36, 37] |
| BCAM0931 | | Monofunctional heme-containing catalase clade 1, homologous to *katA* in *P. aeruginosa* | -0.83 | -0.18 | [5] |
| | | Aerobic respiration | | | |
| BCAL2336 | nuoI | [Fe-S]-dependent NADH dehydrogenase subunit I | n.d. | n.d. | [10, 37] |
| BCAL2343 | nuoB | [Fe-S]-dependent NADH dehydrogenase subunit B | -1.20 | n.s. | [10] |
| BCAL0785 | cydA | Heme-binding cytochrome bd-I ubiquinol oxidase subunit 1 | n.d. | n.d. | [10] |
| BCAL2143 | cyoB | Heme-binding cytochrome bo ubiquinol oxidase subunit 1 | n.s. | n.s. | [38] |
| | | Iron-sulfur cluster formation | | | |
| BCAL2196 | iscA | [Fe-S] assembly accessory protein | n.d. | n.d. | [10, 37] |
| BCAL2198 | iscS | Cysteine desulfurase | n.d. | n.d. | [10, 35, 36] |
| | | Other | | | |
| BCAL3367 | edd | [Fe-S]-dependent phosphogluconate dehydratase | -3.60 | -1.54 | |

*Log2-fold change after addition of 200 μM dipyridyl to a culture at mid-log phase, harvest after 30 min of further incubation, compared to untreated culture. The threshold for reporting statistically significant fold changes was $p \leq 0.05$, n. s. denotes "not significant". The actual p-values from SPSS ANOVA analysis are presented in S4 Table. [Fe-S]: Iron-sulfur cluster, n.d. not determined. Ref.**: references for regulation by RyhB and/or PrrF.

mutant in *P. aeruginosa*, which showed a growth defect compared to wild type under iron depletion [39]. *B. cenocepacia* J2315 is a slow growing small colony variant and possibly less affected by iron starvation. That ΔbrrF is less growth attenuated than WT under iron limitation suggests that key metabolic genes are down-regulated by BrrF.

Overexpressing BrrF from a vector (WT-pBrrF-d1 and WT-pBrrF-d2) attenuated growth (Fig 3B). Overexpressing a derivative of BrrF with point mutations near its 5'end abolished this effect, showing that the growth attenuation depended on the BrrF region which produced the most computationally predicted interactions (Fig 3A).

To test whether growth attenuation by BrrF overexpression was substrate dependent, growth curves were obtained in iron-replete mineral medium supplemented with single carbon compounds. Overexpressing BrrF attenuated growth on nearly all carbon sources tested, including on TCA cycle intermediates such as succinate, citrate, malate and fumarate, on carbohydrates such as glycerol, glucose and gluconate, and on amino acids. Growth of ΔbrrF under iron depletion was tested on five compounds, and it grew faster than WT on four of them (Fig 3C and S4 Fig). This is consistent with a BrrF-dependent downregulation of the TCA cycle, since nearly all growth substrates are either directly or indirectly metabolised through the TCA cycle. The notable exception was propionate, growth on which was not affected by BrrF overexpression under iron-replete conditions, and on which ΔbrrF grew

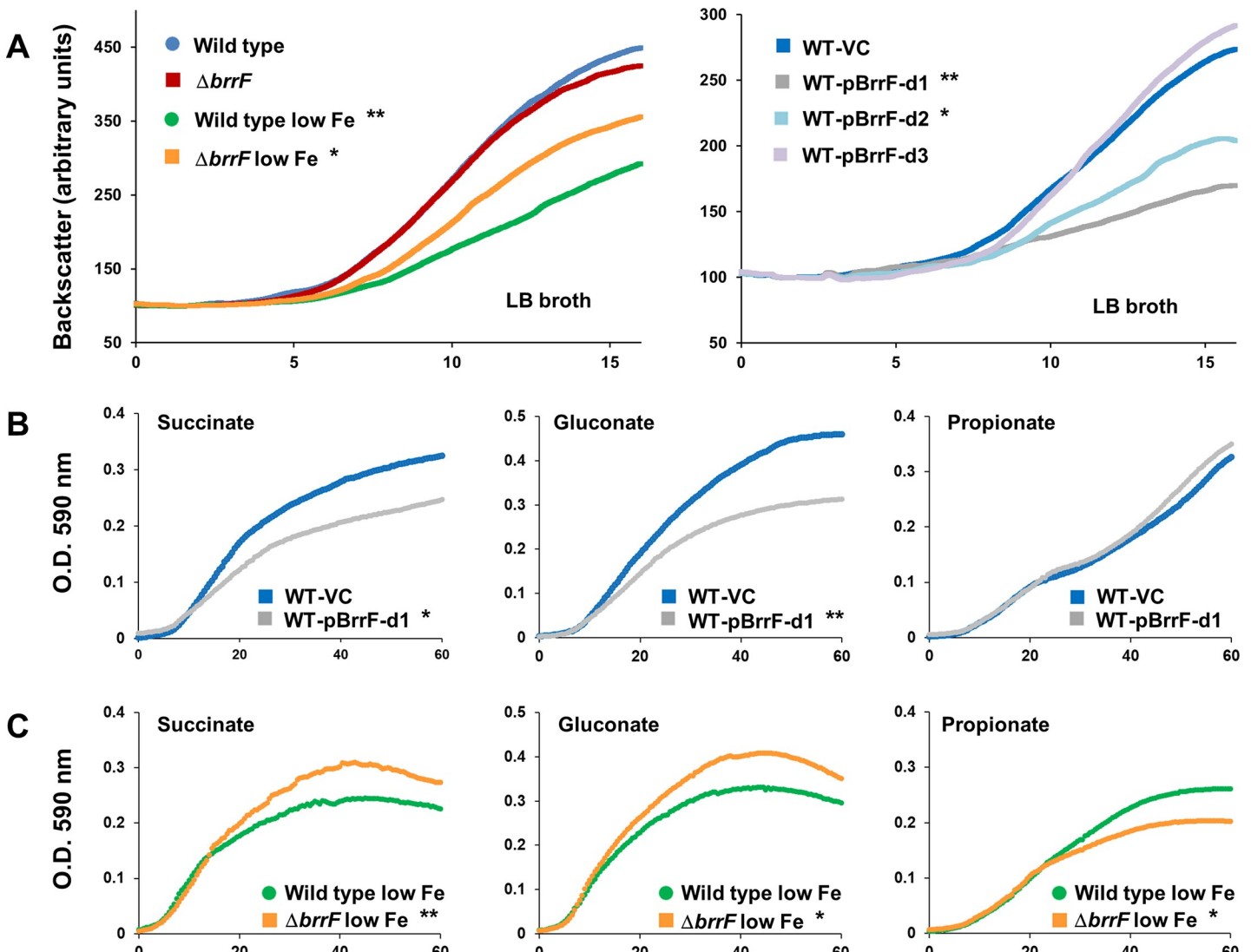

**Fig 3. Growth of *B. cenocepacia* J2313 strains during iron depletion and BrrF overexpression. A)** Left: Wild type and Δ*brrF* were attenuated by iron depletion (addition of 200 µM dipyridyl), Δ*brrF* to a lesser extent than wild type. Right: Overexpressing the native form of BrrF (BrrF-d1 and -d2) under iron-replete conditions attenuated growth. Introducing mutations in the 5' end of BrrF (BrrF-d3) abolished the growth attenuating effect of BrrF overexpression. **B)** On substrates such as succinate and gluconate, overexpressing BrrF attenuated growth. On propionate no attenuation was observed. **C)** Under iron depletion (FeSO₄ omitted from mineral medium), Δ*brrF* grew to a higher O.D. than WT with succinate and gluconate as substrate, while the opposite was true on propionate. Growth was monitored in LB broth in flasks by a Cell Growth Quantifier (A) or in mineral medium in microtiter plates (B, C), the carbon source concentration was 20 mM. Significant differences in growth rate of the wild type or wild type vector control culture and the respective test condition are indicated by asterisks (* = p < 0.05, ** = p < 0.01).

marginally slower than WT under iron depletion. Propionate is metabolised via the methylcitrate cycle in *B. cenocepacia* [40], and the lack of growth attenuation on this substrate suggests that this pathway is less affected by BrrF-dependent down-regulation.

## BrrF regulates the TCA cycle

To test the hypothesis that BrrF induction causes down-regulation of the expression of genes encoding TCA cycle enzymes, all predicted target genes involved in the TCA cycle, and some additional TCA cycle genes which were not predicted targets, were assessed for BrrF-dependent gene expression changes. All samples were collected 30 min after adding rhamnose and/or dipyridyl to cultures initially grown without inducer.

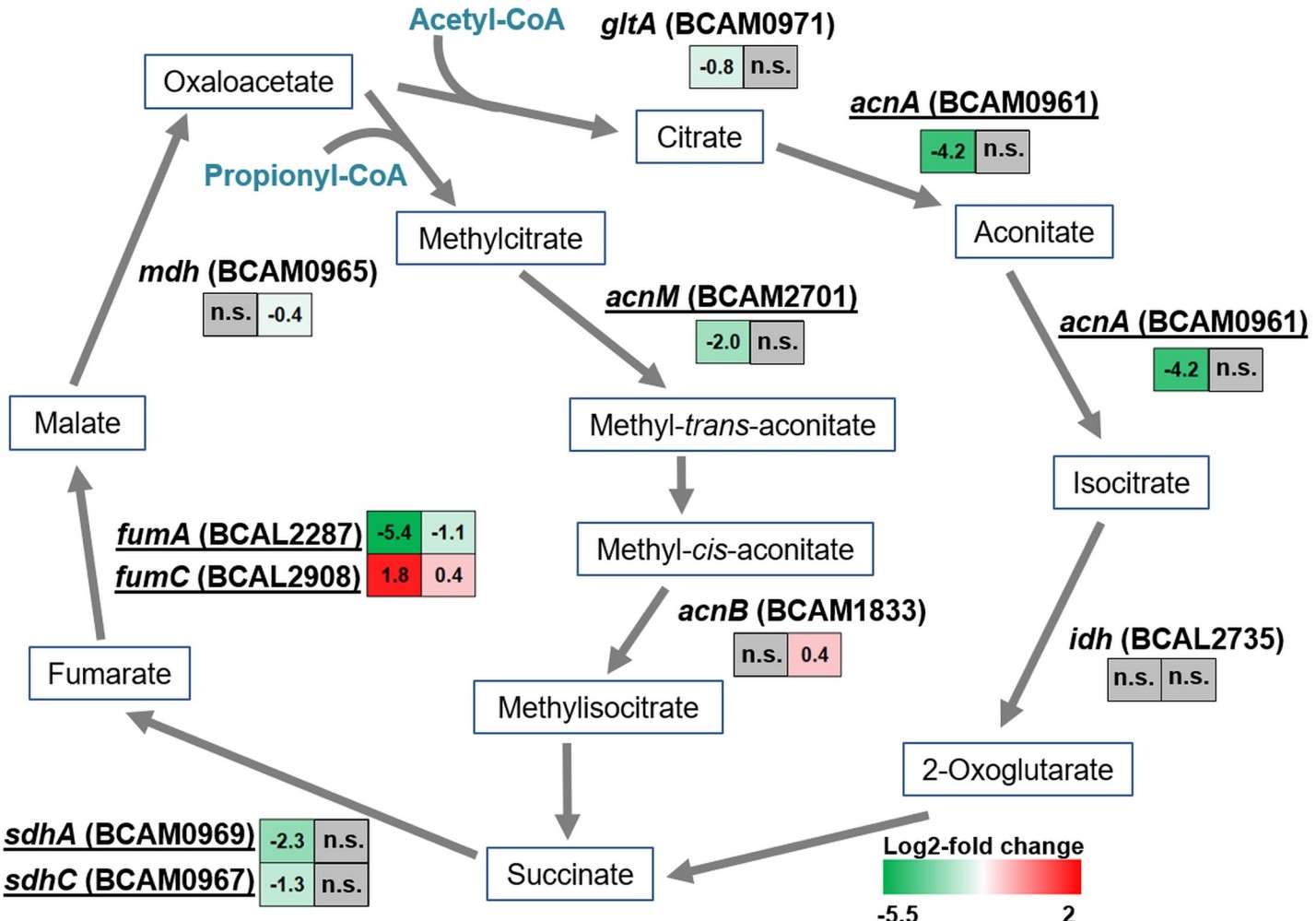

**Fig 4. BrrF-dependent regulation of genes of the tricarboxylic acid and methylcitrate cycle in response to iron depletion.** Iron depletion was invoked by adding dipyridyl to a final concentration of 200 μM to a culture in mid-log phase (LB broth, incubated on a shaker). Cells were then harvested for RNA extraction after 30 min further incubation. This pulse-expression set-up was chosen to minimize secondary effects of iron depletion on gene expression. Predicted targets are underlined. In colored boxes: log2-fold gene expression changes upon iron depletion in WT (left) and ΔbrrF (right). Grey boxes denote no significant changes in expression (i.e. p > 0.05). Bottom right: Color scale for log2-fold changes.

In the WT strain, all TCA genes with complementarity to BrrF were differentially regulated in response to iron depletion. Most of the genes were downregulated, with *acnA* and *fumA* most affected (Table 1, Fig 4). *fumC*, encoding an iron-free fumarate hydratase with an analogous function to [4Fe-4S]-dependent FumA, was upregulated under iron depletion. In ΔbrrF, this regulation was attenuated or abolished (Table 1, Figs 4 and 5). Of the TCA cycle enzymes not predicted as targets, only *gltA* was downregulated significantly in WT under iron depletion. *gltA* is located downstream of and in the same operon as *sdhABCD* [41] and the reduced expression of *gltA* under iron depletion is therefore likely a downstream effect of *sdhC* and *sdhA* downregulation. Of the putative target genes tested, only *fumC* is upregulated under iron depletion, and in a BrrF-dependent manner. In *E. coli fumC* is upregulated under iron depletion via iron-dependent activation by the SoxR protein, probably as an iron-saving measure [42]. In *P. aeruginosa*, expression of *fumC* is Fur-regulated [43], whereas no Fur box is associated with *fumC* in *B. cenocepacia* J2315. The upregulation of *fumC* expression observed in the present study can therefore be a result of BrrF interacting with *fumC* mRNA.

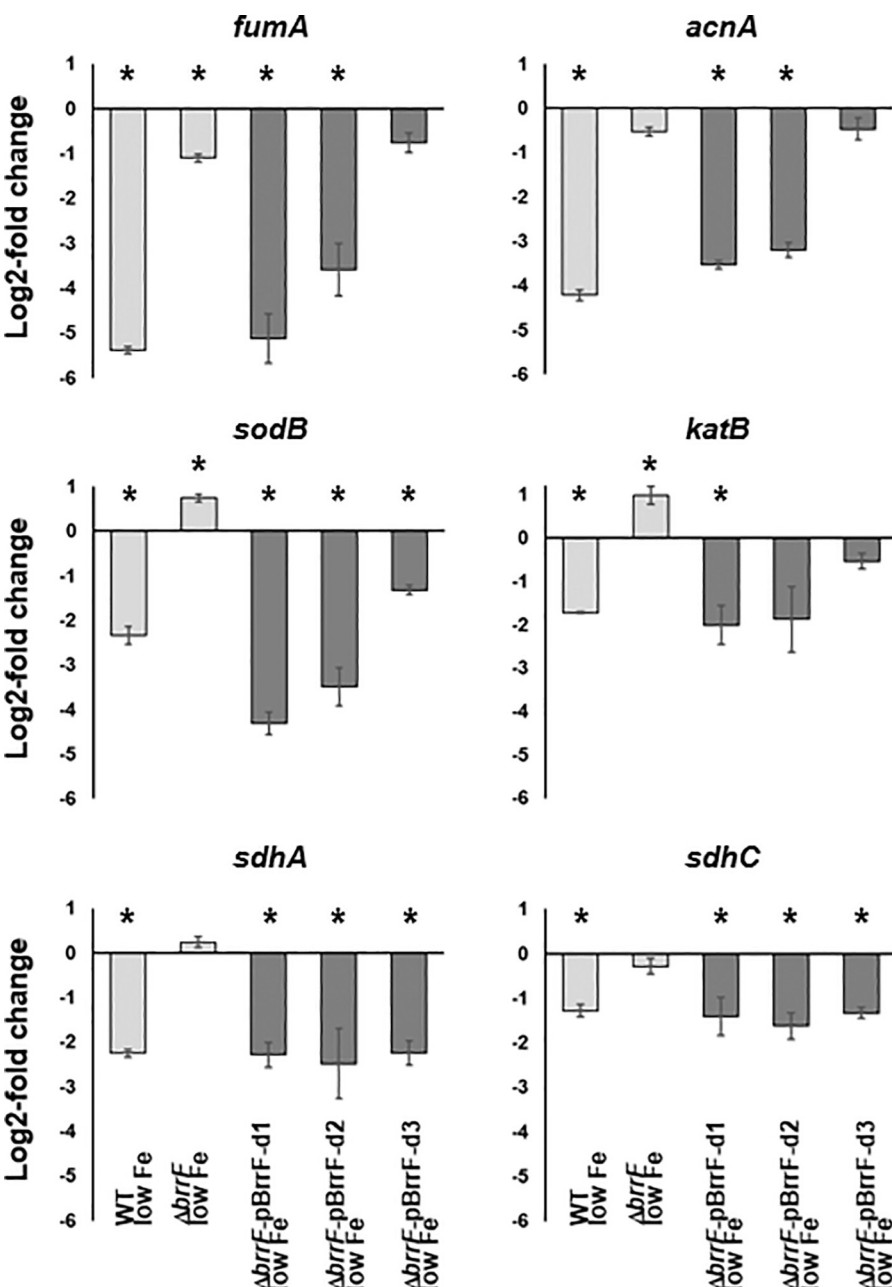

**Fig 5. BrrF-dependent expression of computationally predicted targets under low iron condition.** Cells were harvested for RNA extraction after "pulse-expression" of 30 min in a mid-log culture. Downregulation of gene expression could be complemented in Δ*brrF* by overexpressing BrrF *in trans*. Complementation was dependent on the 5'end of BrrF for most genes except for *sdhA* and *sdhC*. Light grey: Expression compared to iron-replete condition. Dark grey: Expression compared to Δ*brrF*-vector control. pBrrF-d1 and pBrrF-d2: plasmids overexpressing native BrrF with or without processing. pBrrF-d3: derivative of pBrrF-d2 with point mutations near the BrrF 5'end. Asterisks: Significant difference to the respective control, with p ≤ 0.05.

sRNAs capable of both repressing and activating gene expression have been described [44]. The most common mechanisms for activation are reduction of secondary structure in the region of the Shine-Dalgarno sequence, and occluding RNase sensitive sites [45]. BrrF is predicted to interact with fumC mRNA in its 5'UTR, 65 to 75 nt upstream of the gene start.

Secondary structure analysis of the fumC 5'UTR revealed a rho-independent terminator-like hairpin structure at position 32 to 51 from gene start, suggesting transcription termination as a possible mechanism of FumC expression regulation. However, the putative interaction site does not overlap the hairpin structure, and cleaving the 5'UTR at the interaction site does not change the structure of the predicted hairpin. The mechanism of transcription activation by BrrF is therefore not apparent from secondary structure analysis.

Overexpressing BrrF in Δ*brrF* under iron-depletion complemented the downregulation of the tested genes (S4 Table, Fig 5), whereas overexpressing the BrrF derivative with mutations at its 5' end (pBrrF-d3) abolished or reduced that effect, with the notable exception of *sdhC* and *sdhA*. The BrrF region predicted to be important for the interaction with these two genes is further away from the 5' end and is not altered in pBrrF-d3 (Fig 2A and 2B). The abundance of the three BrrF derivatives over-expressed *in trans* was similar (S5 Table).

Under iron-replete conditions, overexpressing BrrF in the wild type strain significantly repressed *acnA* and *fumA*, and induced *fumC* (S4 Table). In line with this, aconitase activity was also reduced by BrrF overexpression (Fig 6). This suggests that growth attenuation during BrrF overexpression under iron-replete conditions (Fig 3B) is mainly due to repression of *acnA*. Expression of *sdhA* and *sdhC* is not affected in this condition, and metabolism of propionate via the methylcitrate cycle completely bypasses AcnA (Fig 4). Fumarate dehydratase activity might overall not be affected by BrrF overexpression, because FumC can replace FumA.

## BrrF regulates the response to oxidative stress

Predicted targets genes *sodB* and *katB* were downregulated significantly in WT under low iron condition (Table 1, Fig 5). SOD activity was significantly reduced in WT cultures under iron limitation, while SOD activity in Δ*brrF* remained unchanged compared to iron-replete condition (Fig 7A). Reduced SOD activity could be complemented by overexpressing the native BrrF in Δ*brrF in trans*, while overexpressing the mutated BrrF derivative did not complement the phenotype. In a $H_2O_2$ sensitivity assay, overexpression of BrrF in Δ*brrF* lead to a significant increase of size of inhibition zone (Fig 7B).

*B. cenocepacia* J2315 has two SODs, the cytoplasmic iron-containing SodB (BCAL2757) and the periplasmic Cu-Zn-containing SodC (BCAL2643) [41, 46]. However, only *sodB* is predicted as target for BrrF, and only *sodB* changed expression significantly (S4 Table). The observed SOD activity changes are therefore probably caused by *sodB* expression changes.

The *B. cenocepacia* J2315 genome encodes four catalases [41], the bifunctional, heme containing catalase/peroxidases KatA (BCAM2107, [20]) and KatB (BCAL3299, [20, 47]), an additional monofunctional heme-containing catalase (BCAM0931) and a manganese-containing catalase (BCAS0635). *katB* and BCAM0931 are predicted as targets for BrrF. *katA* did not change expression upon iron depletion (S4 Table) and BCAS0635 is not expressed under iron-replete conditions [16]. Taken together, this suggest the increased sensitivity to $H_2O_2$ is linked to downregulation of *katB*, which is the major determinant for catalase activity in *B. cenocepacia* [20]. Concomitantly, baseline expression of *katB* is relatively high compared to *katA*, with a 2-fold reduction in the media containing rhamnose and trimethoprim which were used for overexpression experiments (S5 Table). This lower expression of *katB* could be the reason for the larger effect of BrrF overexpression on $H_2O_2$ sensitivity compared to induction of BrrF by low iron (Fig 7), as this would increase the excess of BrrF relative to its target *katB*.

## BrrF-dependent regulation of other predicted targets

While computational target prediction often results in a large number of false positive predicted targets [11], BrrF likely regulates additional genes besides the ones mentioned above.

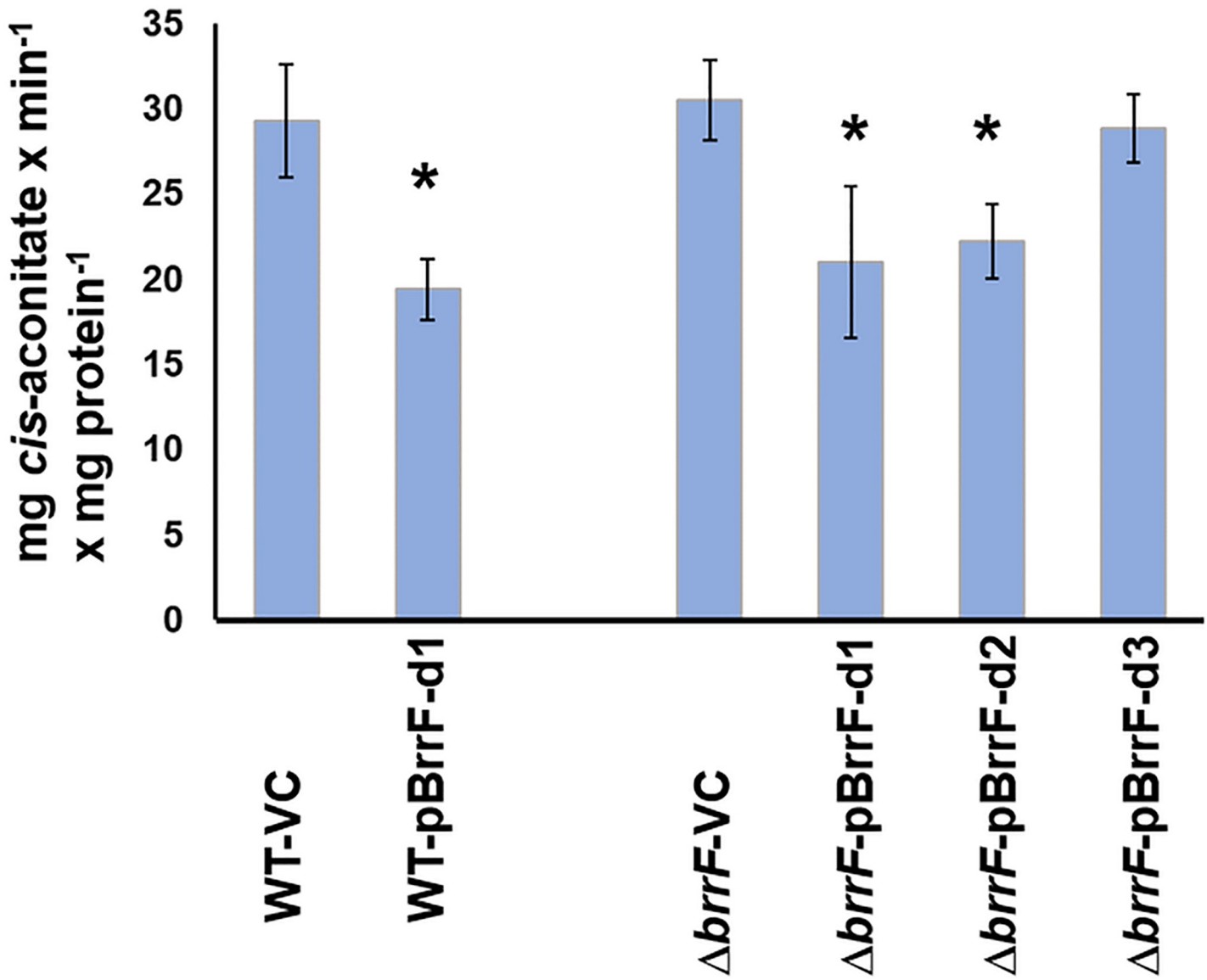

**Fig 6. BrrF-dependent reduction in aconitase activity.** Cultures were grown under iron-replete conditions in LB broth to mid-log phase and rhamnose was added one hour before harvest and protein extraction. Cell-free protein extracts in a 20 mM citrate buffer were supplemented with 40 mM DL-isocitrate and increase in absorbance was measured at 240 nm in a spectrophotometer. Asterisks: Significant difference to the respective control, with $p \leq 0.05$. VC: vector control.

Other predicted targets of BrrF are also involved in metabolism. The nuo respiratory complex is known to be repressed by RyhB in *E. coli* under iron depletion [48], while cytochrome bo oxidase genes *cyoABC* are regulated by RyhB in *Salmonella typhimurium* in response to nitrosative stress [38]. *cyoB* and *nuoB* were tested for BrrF-dependent gene expression changes; *nuoB* was downregulated in WT under iron depletion (Table 1), while *cyoB* did not change expression.

The *edd* gene (BCAL3367; encoding a [Fe-S]-cluster containing phosphogluconate dehydratase) was downregulated in response to iron depletion in a BrrF-dependent manner (Table 1, S4 Table). *edd* is the first gene of the Entner-Doudoroff pathway for carbohydrate degradation, particularly important in *B. cenocepacia* because *Burkholderia* spp. lack a complete glycolysis pathway. Moreover, *edd* is upregulated by $H_2O_2$ stress [23], and has been

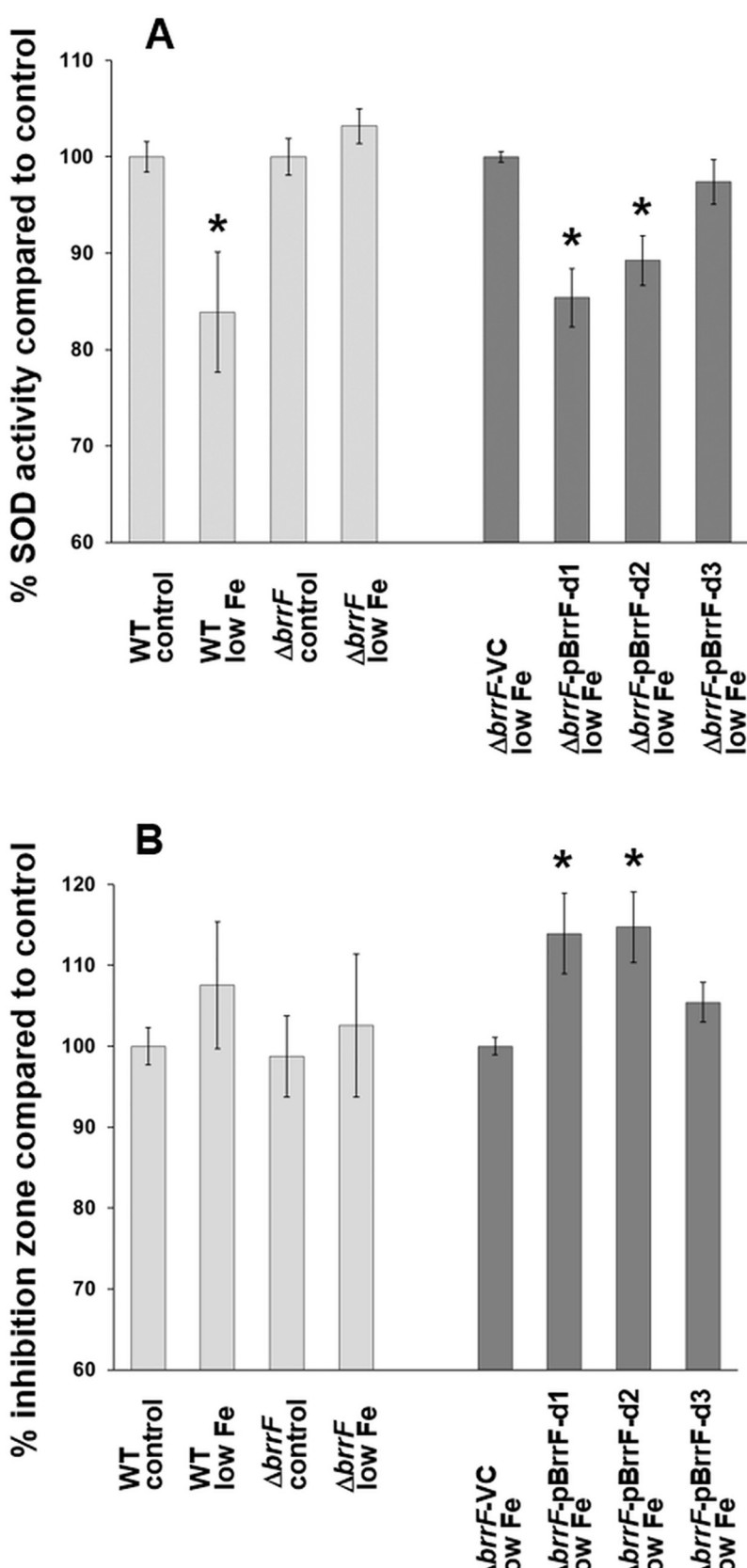

**Fig 7. BrrF-dependent regulation of the oxidative stress response. A)** SOD activity was determined in a cell-free protein extract from planktonic mid-log phase cells, indirectly via superoxide utilization of a xanthine oxidase. Results are presented as percent compared to control condition. **B)** Sensitivity to $H_2O_2$ was determined on low-nutrient agar plates, via formation of a growth inhibition zone around a filter disk containing 1% $H_2O_2$. Plates were pre-incubated for one hour to induce gene expression changes before applying the filter disks. Light grey bars: Response to iron depletion in wild type (WT) and Δ*brrF*. Dark grey bars: Overexpression of native BrrF with or without processing (pBrrF-d1 and pBrrF-d2) or of a derivative with point mutations near the BrrF 5'end (pBrrF-d3). Asterisks: Significant difference to the respective control, with p ≤ 0.05. VC: vector control.

implicated in the oxidative stress response of *Pseudomonas putida* [49]. In *P. putida*, an organism also lacking a complete glycolysis pathway, phosphogluconate dehydratase is necessary for generating NADPH in order to combat oxidative stress. Therefore, downregulating phosphogluconate dehydratase is possibly not only saving iron, but also contributing to overall NADPH balance in the cell.

## Conclusion

Our results demonstrate that sRNA BrrF is involved in downregulating TCA and oxidative stress response in *B. cenocepacia* J2315. This sRNA can be added to the growing list of sRNA involved in regulating metabolism, growth and stress responses in the versatile opportunistic pathogen *B. cenocepacia* [16, 32].

The exact molecular mechanisms of BrrF regulation remain to be resolved. RyhB regulates target genes mainly by binding to its cognate sequence and subsequently increasing the rate of mRNA degradation. This process is Hfq- and RNase E-dependent [50]. It seems likely that BrrF also acts this way, as its sequence is complementary to homologs of confirmed RyhB targets, and most interactions do not include the Shine-Dalgarno sequence of target mRNAs. However, indirect regulation by BrrF via a different yet unknown mechanism cannot be ruled out.

## Supporting information

**S1 Fig. Conservation and secondary structure of BrrF homologues.** The full length of BrrF is very conserved throughout *Burkholderia* (first 8 lines), *Paraburkholderia* (lines 9–16) and *Pandoraea* sp. (lines 17–21). In *Cupriavidus* sp. (lines 22–26) only the first 45 nt of BrrF are present. Underlined are bases conserved in all sequences. Consensus secondary structures were computed using the sequences of the alignment. Red: fully conserved compatible base pairs. Alignment and consensus structures were computed using LocARNA [25].
(TIF)

**S2 Fig. Fur box consensus sequence for *B. cenocepacia* J2315.** The canonical 19 bp palindromic Fur binding site is indicated by a bracket.
(TIF)

**S3 Fig. Density plots of predicted interaction regions.** mRNA regions are depicted in the upper panel and sRNAs regions in the lower panel. The graphs represent all predictions with P< 0.01. x-axis depicts the nucleotide position, position 1 in mRNA is the first nucleotide of the coding sequence. The y-axis depicts the relative frequency of a nucleotide position being part of the predicted sRNA–target interactions.
(TIF)

**S4 Fig. Growth of *B. cenocepacia* J2313 strains during iron depletion and BrrF overexpression.** Growth was monitored in microtiter plates. y-axis: Optical density (590 nm). x-axis:

Time (hours).
(TIF)

**S1 Table. Strains and plasmids used in this study.**
(PDF)

**S2 Table. Primers used in this study.**
(PDF)

**S3 Table. Computationally predicted targets of BrrF.**
(XLSX)

**S4 Table. All log2-fold changes in gene expression determined by qPCR.**
(PDF)

**S5 Table. All qPCR raw Cq values.**
(XLSX)

## Author Contributions

**Conceptualization:** Andrea M. Sass.

**Data curation:** Andrea M. Sass.

**Funding acquisition:** Tom Coenye.

**Investigation:** Andrea M. Sass, Tom Coenye.

**Methodology:** Andrea M. Sass.

**Project administration:** Tom Coenye.

**Resources:** Tom Coenye.

**Software:** Andrea M. Sass.

**Validation:** Andrea M. Sass.

**Visualization:** Andrea M. Sass.

**Writing – original draft:** Andrea M. Sass.

**Writing – review & editing:** Tom Coenye.

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
