## [Decision Letter · Decision Letter 0]

13 Jan 2020

PONE-D-19-34895

Low iron-induced small RNA BrrF regulates central metabolism and oxidative stress responses in *Burkholderia cenocepacia*

PLOS ONE

Dear Dr. Sass,

Two expert reviewers have evaluated your manuscript and their comments are provided below. As you can see, these reviewers appear to have differing opinions about whether or not the data presented support the conclusions drawn in the paper. Other specific points that need to be addressed are also raised by both reviewers. Considering these evaluations, I am going to ask that you submit a revised version of  the manuscript that adequately and appropriately addresses all of  the individual concerns raised by both of  these reviewers.

We would appreciate receiving your revised manuscript by April 13, 2020. To enhance the reproducibility of your results, we recommend that if applicable you deposit your laboratory protocols in protocols.io, where a protocol can be assigned its own identifier (DOI) such that it can be cited independently in the future. For instructions see: http://journals.plos.org/plosone/s/submission-guidelines#loc-laboratory-protocols

We look forward to receiving your revised manuscript!

Sincerely,

R. Martin Roop II, Ph.D.

Academic Editor

PLOS ONE

Journal Requirements:

2. We noted in your submission details that a portion of your manuscript may have been presented or published elsewhere:

'The sRNA has previuosly been reported in a manuscript giving an overview of small RNAs identified in Burkholderia cenocepacia. In that publication, computationally prediction of secondary structure and targets had already been performed, but reported with less detail. These results are included in this manuscript (Fig. 2A and Table S3), with more detail, because they are needed to explain the rationale and the results of the present study. '

Please clarify whether this publicationwas peer-reviewed and formally published.

If this work was previously peer-reviewed and published, in the cover letter please provide the reason that this work does not constitute dual publication and should be included in the current manuscript.

Reviewers' comments:

Reviewer's Responses to Questions

**Comments to the Author**

1. Is the manuscript technically sound, and do the data support the conclusions?

Reviewer #1: Yes

Reviewer #2: Partly

2. Has the statistical analysis been performed appropriately and rigorously? 

Reviewer #1: Yes

Reviewer #2: No

3. Have the authors made all data underlying the findings in their manuscript fully available?

Reviewer #1: Yes

Reviewer #2: Yes

4. Is the manuscript presented in an intelligible fashion and written in standard English?

Reviewer #1: Yes

Reviewer #2: Yes

5. Review Comments to the Author

Reviewer #1: Sass and Coenye describe the characterization of BrrF, an iron regulated sRNA that is conserved in Burkholderia spp. Similar to other iron-regulated sRNAs, BrrF negatively affects the levels of multiple mRNAs encoding iron containing proteins, including those involved the TCA cycle and oxidative stress protection. The authors also describe several very interesting distinctions of BrrF biology compared to previously described iron-regulated sRNAs. Most notably, BrrF is seemingly the result of processing of the 3’ end of the hemP mRNA, which encodes for a transcriptional regulator of heme uptake. The paper is well-written and the conclusions are supported by the results. I do have a couple of issues that I would like the authors to address to more appropriately place this study in the larger body of literature on bacterial iron-regulated sRNAs.

-Line 341-342: BfrB in P. aeruginosa is not regulated by PrrF (see Figure 5 in Wilderman, et al, PNAS 2004). In E. coli, RyhB was suggested to regulate FtnA, not BfrB, but latter studies showed that iron induction of FtnA is similarly independent of RyhB (Nandal, et al, Mol Micro 2010).

-The conclusion section should be expanded to discuss the following points:

1. Figure 3. It seems counter that loss of a gene that is so highly conserved would result in a growth enhancement, especially when the loss of ryhB, PrrF, and other low-iron induced sRNAs results in a growth defect in low iron condition due to loss of iron sparing. This should at the very least be discussed in the concluding section.

2. The authors state that many of the complementarities with BrrF targets do not overlap the SD or start site, which is contrast to other negatively regulated targets of trans-acting sRNAs. Have the authors looked at the structures of any of the target mRNAs to determine how binding by BrrF may affect access to the SD and/or start site?

3. Related, I’m curious if any complementarity was identified in the fumC mRNA - this distinction from how fumC is regulated by iron in other species (directly by Fur versus via the sRNA), is very interesting.

Reviewer #2: The manuscript by Sass and Coenye describes a small regulator RNA (sRNA) in Burkholderia cenocepacia, and the authors have named this sRNA BrrF for Burkholderia regulatory RNA involving iron, Fe. The group previously identified BrrF (called ncS63) as an sRNA that is significantly upregulated in response to low iron conditions, and in this work the authors hypothesized that BrrF is an functional analog of the RyhB sRNA of E. coli that is a well described sRNA involved in iron homeostasis.

In this work, the authors demonstrate that that the brrF gene is likely co-expressed with an upstream gene called hemP that encodes a small protein involved heme iron uptake in other Burkholderia species, and it is predicted that hemP and brrF are regulated by the ferric uptake regulator, Fur. Computational analyses were used to predict regulatory targets of BrrF, and these approaches predicted that BrrF controls the expression of iron-containing enzymes of the TCA cycle and other iron-containing proteins, such as superoxide dismutase and catalase. brrF deletion and over-expression strains were used to assess the regulation of the predicted genes by BrrF, and some of the predicted targets were determined to be authentic targets of BrrF. Subsequent experiments examined the role of BrrF in the response of B. cenocepacia to oxidative stress.

Overall, the work is highly speculative on several of the conclusions that are made, and the authors need to address several important issues, which are outlined below:

-In lines 27-28, the authors state that "BrrF is a Fur-regulated small RNA," however, in lines 296-297, the authors state that "BrrF is therefore probably under the control of the Fur regulator." There is not direct evidence that Fur regulates brrF (or hemP for that matter) in B. cenocepacia. The authors are relying solely on the presence of putative "Fur boxes" for this conclusion. While suggestion that brrF "might" be controlled by Fur is an acceptable proposition, there authors have not presented any direct evidence that brrF is actually Fur-regulated.

-Regarding the deletion and over-expression of brrF, there are several important data that are missing. The authors have relied on qPCR to demonstrate levels of BrrF in all of their strains (Fig. S5), but the authors should employ another method, such as northern blot analysis. For example, Fig. S5 shows that the Cq value for BrrF in the brrF deletion strain in unstressed conditions is approximately 31-32, and the Cq for BrrF in the same strain under low Fe conditions is approximately 25. How can there be such a substantial increase in the levels of BrrF in the brrF deletion strain in different conditions? This shows the unreliable nature of qPCR for analyzing sRNAs. Moreover, given this issue, much of the data in the manuscript are difficult to interpret.

-Also regard the deletion strain, are hemP levels altered by deletion of brrF? If so, it is very difficult to conclude that any phenotypes observed are related only to BrrF.

-Figure 3 should be statistically analyzed. The authors state that strains are "marginally" different, but are these differences statistically significant?

-Lines 453-460 and Figure 7. These data are extremely difficult to understand. To begin with, the authors use confusing terminology: "significant increase of inhibition." This appears to translate to increase sensitivity, but it is hard to know exactly. Regarding the data, WT vs. WT in low Fe shows no difference in sensitivity to oxidative stress, and there should be a >50-fold increase in BrrF in the WT-low Fe condition compared to WT based on the authors previous work (Ref. 9). Additionally, the brrF deletion strain in both conditions has similar sensitivity levels to those of the WT and WT-low Fe. However, when you over-express versions of brrF in the brrF deletion strain, these strains exhibit increased sensitivity to oxidative stress. Is there a greater than 50-fold increase in BrrF in these strains? Can over-expression of these brrF genes in the WT strain similarly increase sensitivity to oxidative stress. The authors need to carefully evaluate the results from these experiments.

6. PLOS authors have the option to publish the peer review history of their article (what does this mean?). If published, this will include your full peer review and any attached files.

Reviewer #1: No

Reviewer #2: No

---

## [Author Response · Author response to Decision Letter 0]

9 Apr 2020

Manuscript “ Low iron-induced small RNA BrrF regulates central metabolism and oxidative stress responses in Burkholderia cenocepacia” by Sass and Coenye

Response to reviewers’ questions.

Reviewer 1:

-Line 341-342: BfrB in P. aeruginosa is not regulated by PrrF (see Figure 5 in Wilderman, et al, PNAS 2004). In E. coli, RyhB was suggested to regulate FtnA, not BfrB, but latter studies showed that iron induction of FtnA is similarly independent of RyhB (Nandal, et al, Mol Micro 2010).

Answer:

The reviewer is correct, this was an oversight on our side and we deleted the sentence from the discussion. 

-The conclusion section should be expanded to discuss the following points:

1. Figure 3. It seems counter that loss of a gene that is so highly conserved would result in a growth enhancement, especially when the loss of ryhB, PrrF, and other low-iron induced sRNAs results in a growth defect in low iron condition due to loss of iron sparing. This should at the very least be discussed in the concluding section.

Answer:

The B. cenocepacia strain used in this study, J2315, is a slow growing small colony variant. Strain J2315 permanently displays a growth deficiency compared to other B. cenocepacia laboratory strains such as K56-2 and H111. It is therefore probably less affected by iron starvation and can maintain growth for longer under those conditions compared to “normally” growing bacteria. Absence of downregulation of key metabolic genes, as in �brrF, can then increase growth, and not be detrimental to the survival of the bacteria. 

We have added this rationale to the manuscript main text (lines 370-375: “This is in contrast to observations with a PrrF double deletion mutant in P. aeruginosa, which showed a growth defect compared to wild type under iron depletion [39]. B. cenocepacia J2315 is a slow growing small colony variant and possibly less affected by iron starvation. That �brrF is less growth attenuated than WT under iron limitation suggests that key metabolic genes are down-regulated by BrrF.”).

2. The authors state that many of the complementarities with BrrF targets do not overlap the SD or start site, which is contrast to other negatively regulated targets of trans-acting sRNAs. Have the authors looked at the structures of any of the target mRNAs to determine how binding by BrrF may affect access to the SD and/or start site?

Answer:

We have performed secondary structure analysis for the genes listed in Table 1, including the 5’UTR to max. 200 nt upstream of the gene start and to 100 nt downstream of the gene start. The putative interaction region was not involved in occluding the SD or the gene start, and repeating the analysis after virtual cleavage of the mRNA at the predicted interaction site did not reveal occlusion of SD or gene start after cleavage. Computational secondary structure analysis was therefore not informative regarding mechanism of action of BrrF.

We have added the findings from secondary structure analysis to the manuscript text (lines 330-332: “Secondary structure analysis of putative target mRNAs before and after virtual cleavage at the predicted interaction site did not indicate an occlusion of Shine-Dalgarno sequences by mRNA intramolecular structures.”).

3. Related, I’m curious if any complementarity was identified in the fumC mRNA - this distinction from how fumC is regulated by iron in other species (directly by Fur versus via the sRNA), is very interesting.

Answer:

Activation of expression by a small RNA is less common than downregulation. The most common mechanisms for activation are reduction of secondary structure in the region of the SD sequence, and occluding RNase sensitive sites. 

Secondary structure analysis of the 5’UTR of fumC mRNA showed that it contains a hairpin formation at position 32 to 51 upstream of the gene start, which could act as a rho-independent terminator. The SD or gene start are not occluded by secondary structure formation according to this analysis. The binding site of BrrF is predicted for position 65 to 75, not overlapping the putative terminator structure. We repeated the secondary structure analysis after removing the RNA sequence upstream of and including the interaction site, and the terminator structure still formed unchanged. It is therefore not apparent from structure analysis what the exact mechanism of activation is. 

We added the special characteristics of the fumC 5’UTR to the manuscript text (lines 431-441: “sRNAs capable of both repressing and activating gene expression have been described [44]. The most common mechanisms for activation are reduction of secondary structure in the region of the SD sequence, and occluding RNase sensitive sites [45]. BrrF is predicted to interact with fumC mRNA in its 5’UTR, 65 to 75 nt upstream of the gene start. Secondary structure analysis of the fumC 5’UTR revealed a rho-independent terminator-like hairpin structure at position 32 to 51 from gene start, suggesting transcription termination as a possible mechanism of FumC expression regulation. However, the putative interaction site does not overlap the hairpin structure, and cleaving the 5’UTR at the interaction site does not change the structure of the predicted hairpin. The mechanism of transcription activation by BrrF is therefore not apparent from secondary structure analysis.”). 

Reviewer 2:

-In lines 27-28, the authors state that "BrrF is a Fur-regulated small RNA," however, in lines 296-297, the authors state that "BrrF is therefore probably under the control of the Fur regulator." There is not direct evidence that Fur regulates brrF (or hemP for that matter) in B. cenocepacia. The authors are relying solely on the presence of putative "Fur boxes" for this conclusion. While suggestion that brrF "might" be controlled by Fur is an acceptable proposition, there authors have not presented any direct evidence that brrF is actually Fur-regulated.

Answer:

In the particular B. cenocepacia strain used for this study, Fur is an essential gene (reference Wong et al., 2016, Candidate essential genes in Burkholderia cenocepacia J2315 identified by genome-wide TraDIS. Front. Microbiol. 7:1288. doi: 10.3389/fmicb.2016.01288). In other Burkholderia strains, Fur can be deleted, and it has been deleted in B. multivorans, also a member of the Burkholderia cepacia complex and relatively closely related to B. cenocepacia. The Fur-dependent expression of the HemP protein upstream of BrrF has been demonstrated for that bacterium (ref. 21). The sequences upstream of hemP are identical in B. cenocepacia J2315 and the B. multivorans strain used in that study, including the Fur box. The sequence upstream of hemP of B. multivorans has been used in a Fur titration assay in that study (ref. 21), showing that E. coli Fur protein can bind to this Fur box sequence. The same sequence was also bound by B. multivorans Fur in an EMSA experiment (ref. 21). It is therefore very likely that the hemP upstream sequence in B. cenocepacia is also bound by Fur. 

We have added these observations to the manuscript text, lines 302-307 (“The sequence upstream of the hemP gene start in B. cenocepacia is identical to that upstream of hemP in B. multivorans for the first 77 nt, which includes the Fur box. This B. multivorans sequence was bound by B. multivorans Fur protein in an electrophoresis mobility shift assay [21], the respective sequence in B. cenocepacia is therefore most likely also bound by Fur.”)

-Regarding the deletion and over-expression of brrF, there are several important data that are missing. The authors have relied on qPCR to demonstrate levels of BrrF in all of their strains (Fig. S5), but the authors should employ another method, such as northern blot analysis. For example, Fig. S5 shows that the Cq value for BrrF in the brrF deletion strain in unstressed conditions is approximately 31-32, and the Cq for BrrF in the same strain under low Fe conditions is approximately 25. How can there be such a substantial increase in the levels of BrrF in the brrF deletion strain in different conditions? This shows the unreliable nature of qPCR for analyzing sRNAs. Moreover, given this issue, much of the data in the manuscript are difficult to interpret.

Answer:

The reason to show the qPCR raw data was to point out that the levels of BrrF in overexpression experiments under low iron conditions are comparable to the levels of BrrF in wild type under low iron conditions. In the brrF deletion mutant, Cq values for BrrF are between 25 and 32, which is indeed relatively high, given that BrrF should not have been detected. However, our no-RT controls showed that primers for BrrF always give signals in this order of magnitude, as Cq values for BrrF in no-RT controls ranged from 32 to 27. In contrast, no-RT control Cq values for all other genes investigated were always larger than 34. It appears that, for reasons unknown we had measurable brrF signals that were larger than for all the other genes investigated. However, whatever the reason for this high background signals, we are convinced they have no influence on our conclusions. After all, the Cq values for BrrF in wild type under iron replete conditions are approx. 15, while those under conditions where BrrF is induced are approx. 10-11. This difference between Cq values of 10-15 on one hand, and Cq values of 25-32 on the other hand, is large enough. Generally, no-RT and no-template control Cq values should be >5 cycles larger than sample Cq values, to contribute <3% to the signal (Nolan et al., 2006, Quantification of mRNA using real-time RT-PCR. Nat. Meth. 1:1559, doi:10.1038/nprot.2006.236), and this is clearly the case for our data . Because of this we believe our qPCR data are reliable and we decided not to quantify BrrF expression in another way.

We have supplied all qPCR raw Cq values, including no-RT controls, as supplementary material, in Table S5, and have removed Figure S5, since it depicts data which is now available in Table S5.

-Also regard the deletion strain, are hemP levels altered by deletion of brrF? If so, it is very difficult to conclude that any phenotypes observed are related only to BrrF.

Answer:

hemP mRNA levels were shown in Figure S5 on the right hand side, they are not altered by brrF deletion. We have now supplied all qPCR raw data in Table S5. 

We conducted additional growth curves to assess the functionality of HemP. HemP is necessary for heme uptake in B. multivorans (ref. 29); and our results showed that a brrF deletion mutant was not impaired in heme utilization (Figure S4). It is therefore reasonable to assume that HemP expression is not altered by brrF deletion, and that the observed phenotypes of �brrF are not related to hemP. 

We have added the results of the additional growth tests to the main text (lines 368-369: “�brrF was also not impaired in heme utilisation, indicating that expression of HemP is not affected by brrF deletion (Fig. S4).”), and added representative growth curves, showing the effect of heme addition to the low iron medium, to Figure S4 in supplementary material.

-Figure 3 should be statistically analyzed. The authors state that strains are "marginally" different, but are these differences statistically significant?

Answer:

To perform the statistical analysis for all available data as consistently as possible, we analyzed the growth rates. Part of the growth curves in Figure 3 were obtained with a Cell Growth Quantifier, which runs with CGQuant software with automated growth rate calculation. The equation used by the CGQuant software to calculate time averaged growth rates was applied to the output of the microplate reader. The resulting growth rates were then statistical analyzed. The method section has been amended accordingly (lines 140-144: “Growth rates were determined using the equation µ * h-1 = ln (xt2/xt1)/(t2-t1), where x denotes optical density or backscatter arbitrary units, and t1 and t2 refer to a point at the beginning or end of the analysed time interval, respectively [17]. The obtained values for µ of the biological replicates were then analysed either by One-way ANOVA or a two-tailed Student’s t-test using SPSS (v. 25).”), as well as the legend to Figure 3 (lines 392-394: “Significant differences in growth rate of the wild type or wild type vector control culture and the respective test condition are indicated by asterisks (* = p < 0.05, ** = p < 0.01).“). Statistically significant different growth rates are indicated in Figure 3.

-Lines 453-460 and Figure 7. These data are extremely difficult to understand. To begin with, the authors use confusing terminology: "significant increase of inhibition." This appears to translate to increase sensitivity, but it is hard to know exactly. 

Answer:

We have changed the wording of this sentence from “significant increase of inhibition” to “significant increase in size of inhibition zone” (line 493) to clarify.

-Regarding the data, WT vs. WT in low Fe shows no difference in sensitivity to oxidative stress, and there should be a >50-fold increase in BrrF in the WT-low Fe condition compared to WT based on the authors previous work (Ref. 9). Additionally, the brrF deletion strain in both conditions has similar sensitivity levels to those of the WT and WT-low Fe. However, when you over-express versions of brrF in the brrF deletion strain, these strains exhibit increased sensitivity to oxidative stress. Is there a greater than 50-fold increase in BrrF in these strains? Can over-expression of these brrF genes in the WT strain similarly increase sensitivity to oxidative stress. The authors need to carefully evaluate the results from these experiments.

Answer:

The variations of the effect of BrrF induction on H2O2 sensitivity in WT under low iron (not significant) and in �brrF with BrrF expression complemented under low iron condition (significant) is probably not due to a difference in BrrF expression in these two strains as the reviewer suggested, but due to a reduction of katB expression under the condition of BrrF overexpression. katB is approx. 2-fold lower expressed in media containing rhamnose and trimethoprim, which are used for overexpression experiments. This increases the excess of BrrF in comparison to its target katB mRNA, and could cause an increase in katB down-regulation, which is also apparent in katB Cq values under this condition. 

We have supplied all the qPCR raw data (Table S5), and added the above explanation to the manuscript text (lines 519-524: “Concomitantly, baseline expression of katB is relatively high compared to katA, with a 2-fold reduction in the media containing rhamnose and trimethoprim which were used for overexpression experiments (Table S5). This lower expression of katB could be the reason for the larger effect of BrrF overexpression on H2O2 sensitivity compared to induction of BrrF by low iron (Fig 7), as this would increase the excess of BrrF relative to its target katB.”).

---

## [Decision Letter · Decision Letter 1]

13 May 2020

PONE-D-19-34895R1

Low iron-induced small RNA BrrF regulates central metabolism and oxidative stress responses in *Burkholderia cenocepacia*

PLOS ONE

Dear Dr. Sass,

Both reviewers have pointed out that apparently it has not been experimentally established that Fur regulates the iron responsive expression of * brrF* in *Burkholderia cenocepacia*. If  this is correct, then these reviewers are correct that this point needs to be clarified. Other than clarification of  this important point, both reviewers agree that the manuscript presents important findings. Consequently, I am going to ask that you submit a revised version of  the paper that directly addresses this issue and once this point is clarified, I will accept the manuscript.

We would appreciate receiving your revised manuscript by July 15, 2020. To enhance the reproducibility of your results, we recommend that if applicable you deposit your laboratory protocols in protocols.io, where a protocol can be assigned its own identifier (DOI) such that it can be cited independently in the future. For instructions see: http://journals.plos.org/plosone/s/submission-guidelines#loc-laboratory-protocols

We look forward to receiving your revised manuscript!

Sincerely,

R. Martin Roop II, Ph.D.

Academic Editor

PLOS ONE

Reviewers' comments:

Reviewer's Responses to Questions

**Comments to the Author**

1. If the authors have adequately addressed your comments raised in a previous round of review and you feel that this manuscript is now acceptable for publication, you may indicate that here to bypass the “Comments to the Author” section, enter your conflict of interest statement in the “Confidential to Editor” section, and submit your "Accept" recommendation.

Reviewer #1: (No Response)

Reviewer #2: All comments have been addressed

2. Is the manuscript technically sound, and do the data support the conclusions?

Reviewer #1: Yes

Reviewer #2: Yes

3. Has the statistical analysis been performed appropriately and rigorously? 

Reviewer #1: Yes

Reviewer #2: Yes

4. Have the authors made all data underlying the findings in their manuscript fully available?

Reviewer #1: Yes

Reviewer #2: Yes

5. Is the manuscript presented in an intelligible fashion and written in standard English?

Reviewer #1: Yes

Reviewer #2: Yes

6. Review Comments to the Author

Reviewer #1: Almost all comments have been addressed adequately. I have only one remaining issue. In the abstract the authors state that BrrF is a "Fur-regulated small RNA" (line 27), but as indicated by the other reviewer in the first review Fur-regulation of BrrF is not directly determined for B. ceenocepacia BrrF. This is a reasonable assumption as all data are consistent with this model, but the language in the abstract needs to be modified to remove the certainty.

Reviewer #2: The authors have adequately addressed the comments from the reviewers, and overall, the work is sound.

However, I would like to encourage the authors to seriously assess their use of qRT-PCR for sRNA levels. I am still not entirely convinced that the qRT-PCR data is accurately depicting the levels of BrrF under different conditions and in different strains. The rebuttal from the authors includes the comment that "our no-RT controls showed that primers for BrrF

always give signals in this order of magnitude." Does this not concern you that the primers you are using are not efficient and/or not reliable? I really am trying to be helpful, as I just want to make sure that the data are as accurate as possible. I would suggest that the authors consider employing northern blot analyses as they continue to work in the area of sRNAs. This will only serve to complement your qRT-PCR results, and it will significantly enhance your ability to analyze smaller differences in sRNA levels.

7. PLOS authors have the option to publish the peer review history of their article (what does this mean?). If published, this will include your full peer review and any attached files.

Reviewer #1: No

Reviewer #2: No

---

## [Author Response · Author response to Decision Letter 1]

1 Jul 2020

Reviewer #1: Almost all comments have been addressed adequately. I have only one remaining issue. In the abstract the authors state that BrrF is a "Fur-regulated small RNA" (line 27), but as indicated by the other reviewer in the first review Fur-regulation of BrrF is not directly determined for B. ceenocepacia BrrF. This is a reasonable assumption as all data are consistent with this model, but the language in the abstract needs to be modified to remove the certainty.

Response:

We have changed the wording of the abstract to “BrrF is a small RNA highly upregulated in Burkholderia cenocepacia under conditions of iron depletion and with a genome context consistent with Fur regulation.” (lines 27-28). We also softened the related statement in lines 307-8 in the main manuscript by changing the wording. 

Reviewer #2: The authors have adequately addressed the comments from the reviewers, and overall, the work is sound.

However, I would like to encourage the authors to seriously assess their use of qRT-PCR for sRNA levels. I am still not entirely convinced that the qRT-PCR data is accurately depicting the levels of BrrF under different conditions and in different strains. The rebuttal from the authors includes the comment that "our no-RT controls showed that primers for BrrF always give signals in this order of magnitude." Does this not concern you that the primers you are using are not efficient and/or not reliable? I really am trying to be helpful, as I just want to make sure that the data are as accurate as possible. I would suggest that the authors consider employing northern blot analyses as they continue to work in the area of sRNAs. This will only serve to complement your qRT-PCR results, and it will significantly enhance your ability to analyze smaller differences in sRNA levels.

Response:

We performed Northern blotting for a previous publication (DOI:10.1038/s41598-017-15818-3), using DIG labelled probes and eventually exposing our hybridised membranes to X-ray film. Our lab does not have the setup nor permission for working with radioactively labelled probes. Accurate quantification of the expression levels of BrrF is to our knowledge not possible with that method, because black lines on exposed X-ray film would have to be analysed (please refer to the supplementary material of the publication https://static-content.springer.com/esm/art%3A10.1038%2Fs41598-017-15818-3/MediaObjects/41598_2017_15818_MOESM1_ESM.pdf). Northern blotting with this method will probaby be no improvement on qPCR. On the other hand, qPCR, although not perfect, is to our knowledge sufficient to show overexpression of sRNA. 

The exact level of overexpression of BrrF from the vectors is dependent on the rhamnose concentration in the medium, and is by definition very similar for all constructs. Exactly replicating the BrrF expression level of wild type cells under iron depletion is not necessary for the purpose of showing that complementation does happen. The purpose of the qPCR experiments in this manuscript is monitoring the changes in levels of putative target mRNAs, and prove that those levels are declining upon BrrF overexpression. The exact fold changes were not subject of further interpretation. 

We therefore feel that the additional work and cost involved in setting up Northern blotting experiments is not justified.

---

## [Editor Report · Decision Letter 2]

8 Jul 2020

Low iron-induced small RNA BrrF regulates central metabolism and oxidative stress responses in *Burkholderia cenocepacia*

PONE-D-19-34895R2

Dear Dr. Sass,

We’re pleased to inform you that your manuscript has been judged scientifically suitable for publication and will be formally accepted for publication once it meets all outstanding technical requirements.

Sincerely,

R. Martin Roop II, Ph.D.

Academic Editor

PLOS ONE
---

## [Editor Report · Acceptance letter]

10 Jul 2020

PONE-D-19-34895R2 

Low iron-induced small RNA BrrF regulates central metabolism and oxidative stress responses in *Burkholderia cenocepacia*

Dear Dr. Sass:

I'm pleased to inform you that your manuscript has been deemed suitable for publication in PLOS ONE. Congratulations! Your manuscript is now with our production department. 

Kind regards, 

on behalf of

Dr. Roy Martin Roop II 

Academic Editor

PLOS ONE